# Semantic-Aware Prefix Learning via Token Truncation for Efficient Image Generation

## Abstract

Visual tokenizers play a central role in latent image generation by bridging high-dimensional images and tractable generative modeling. However, most existing tokenizers are still trained with reconstruction-dominated objectives, which often yield latent representations that are only weakly grounded in high-level semantics. Recent approaches improve semantic alignment, but typically treat semantic signals as auxiliary regularization rather than making them functionally necessary for representation learning. We propose `SMAP`, a **S**e**M**antic-**A**ware **P**refix tokenizer that injects semantic conditions as prefix-preserved invariants into a query-based 1D tokenization framework. To make semantics indispensable during training, `SMAP` introduces a *tail token dropping* strategy, which forces semantic conditions and early latent prefixes to bear increasing responsibility via progressive token truncation. This leads to *information-ordered* token sequences that support length-adaptive encoding and graceful truncation. To exploit the resulting latent space for generation, we further introduce `CARD`, a hybrid **C**ausal **A**uto**R**egressive–**D**iffusion generator. `CARD` first models global structural dependencies autoregressively and then refines the conditional distribution via flow matching for high-fidelity synthesis. Extensive experiments on ImageNet show that `SMAP` consistently improves reconstruction quality across discrete and continuous tokenization settings, and that its semantically grounded latent space yields strong downstream generation performance with compact token budgets.

## 1 Introduction

In recent years, image generation has achieved substantial progress across multiple modeling paradigms, including diffusion models (Rombach et al., 2022a; Yao et al., 2024; Ma et al., 2024), autoregressive visual models (Esser et al., 2021; Li et al., 2024a; Tian et al., 2024), and masked generative approaches (Chang et al., 2022; Li et al., 2023b). Despite differences in their generative mechanisms, these methods share a common architectural principle: images are first mapped from the high-dimensional pixel space into a compact latent representation through a learned image encoder or tokenizer (Rombach et al., 2022a; Esser et al., 2021; Yu et al., 2022b). This latent space, which may be continuous or discrete, aims to preserve essential semantic and structural information while significantly reducing dimensionality. Existing research has largely focused on improving the generative stage through advances in model architectures (Peebles & Xie, 2023) and training objectives (Ma et al., 2024), while the role of the latent representation learning mechanism remains comparatively underexplored. However, the structure, expressiveness, and inductive biases of the latent space critically determine both the efficiency and the performance ceiling of downstream generative models (Team et al., 2025; Ke & Xue, 2025), underscoring the importance of systematically studying and improving image encoding and tokenization strategies.

Although an increasing body of work has recognized the importance of latent space quality (Kingma & Welling, 2014b; Tschannen et al., 2025; Rombach et al., 2022a), most existing approaches (Yu et al., 2022a; Zhu et al., 2023; Yu et al., 2024a) still train visual tokenizers using reconstruction-dominated objectives. Their learned latent or token space may exhibit only weak alignment with high-level concepts, limiting its effectiveness as an interface for downstream generative models and impairing semantic controllability.

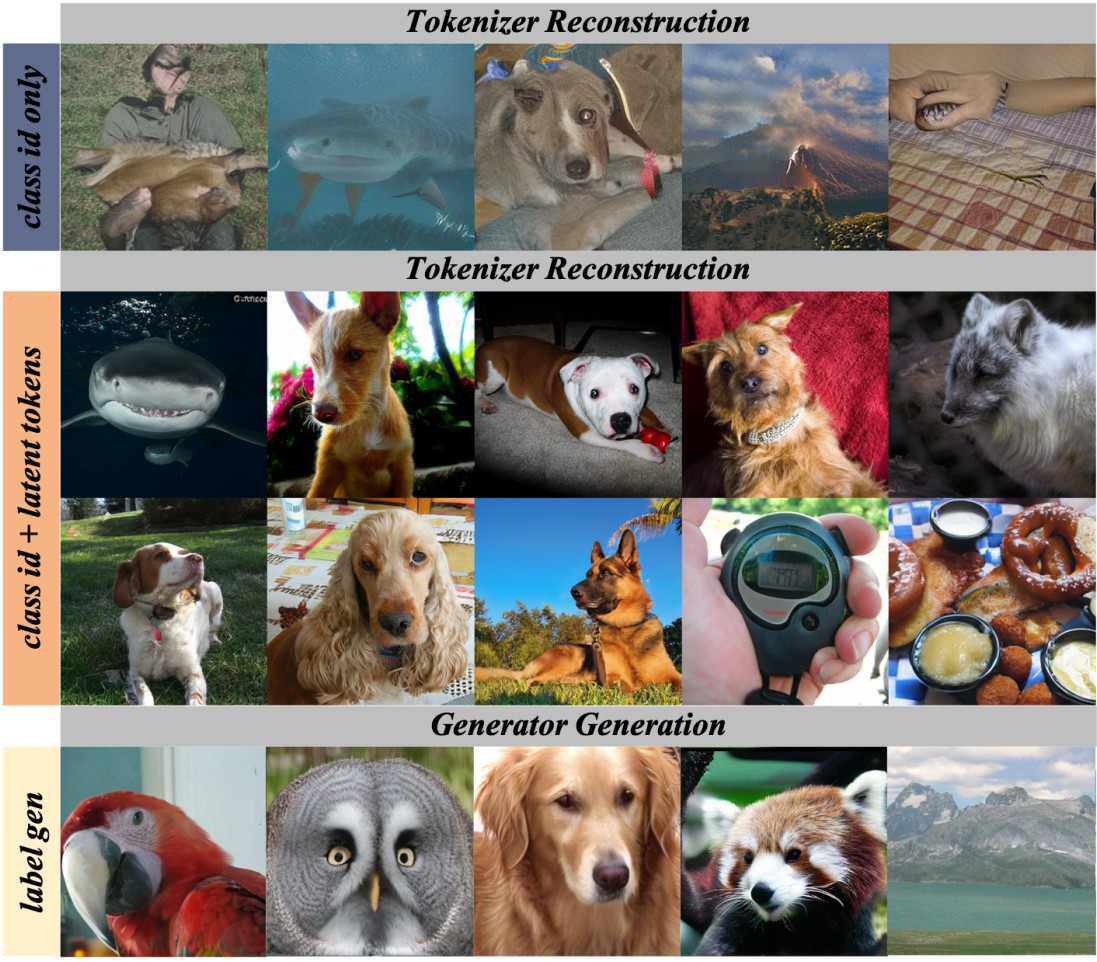

Figure 1: **Semantic-aware prefix learning in reconstruction and generation. Top:** Using only the class condition, `SMAP` reconstructs images that already capture category-level semantics and coarse global structure. **Middle:** Adding latent tokens substantially improves reconstruction fidelity and restores instance-specific details, showing that semantic conditions and latent prefixes play complementary roles. **Bottom:** Based on the resulting semantically grounded token space, `CARD` generates high-quality class-conditional images.

To address this mismatch, recent studies have begun to introduce semantic inductive biases into tokenizer pretraining through alignment or regularization strategies (Yu et al., 2024b; Kim et al., 2025; Jingfeng et al., 2025). A common approach leverages pretrained semantic encoders, such as `CLIP` (Radford et al., 2021), or representation alignment signals (e.g., `REPA` (Yu et al., 2025b)), to encourage correlation between latent codes and high-level semantics. Along similar lines, Visual Tokenizer Pretraining (`VTP` (Jingfeng et al., 2025)) observes that reconstruction-centric training objectives bias token representations toward low-level visual information and struggle to yield concise semantic abstractions, and consequently advocates injecting semantic signals during tokenizer learning.

More importantly, semantic alignment (Chen et al., 2025; Yu et al., 2025b) alone does not guarantee that the token space can carry and express the high-level structural information required for image generation. Many existing approaches merely encourage token representations to be correlated with semantic features in a loose sense, without explicitly requiring semantic signals to bear essential informational responsibility during reconstruction and representation learning. We therefore argue that *the core challenge lies not in whether semantics are aligned, but in how semantic information is made an indispensable component of tokenizer*

*pretraining—such that global structure and high-level concepts are encoded into usable and transferable token representations.*

To this end, we propose `SMAP`, a semantically aware image tokenizer that encodes high-level semantics as prefix-preserved invariants. By construction, semantic information actively participates in both reconstruction and representation learning throughout pretraining, explicitly driving the tokenizer to encode global structure and high-level concepts. `SMAP` employs a query-based 1D tokenizer architecture and a principled tail token dropping strategy to learn information-ordered token sequences, enabling length-adaptive representations with strong semantic grounding. We refer to this behavior as semantic-aware prefix learning: semantic conditions encode high-level identity in the prefix, while later latent tokens progressively refine instance-level detail.

Building upon `SMAP`, we further propose `CARD`, a class of hybrid autoregressive–diffusion generative models for image generation. Following the staged generation principle of `MAR` (Li et al., 2024a), `CARD` decomposes image generation into two complementary components: an autoregressive module that models high-level structural dependencies in the latent space, followed by a `Flow Matching` (Lipman et al., 2023)–based continuous density model that captures and refines the conditional distribution, enabling high-quality image synthesis.

In brief, our contributions are threefold.

- We identify a central limitation of existing tokenizer training pipelines: semantics are typically encouraged through loose alignment objectives, rather than being made functionally necessary for reconstruction and representation learning.

- We propose `SMAP`, a semantic-aware 1D tokenizer that incorporates semantic conditions as prefix-preserved invariants and enforces semantic dependency through token truncation, resulting in semantically grounded, information-ordered, and length-adaptive token sequences.

- We develop `CARD`, a hybrid autoregressive–diffusion generator built on top of `SMAP`, and demonstrate that semantically grounded tokenization consistently improves both tokenizer reconstruction and downstream image generation under compact token budgets.

## 2 Related Work

**Image Tokenization.** Modern generative image models rely critically on image tokenization to enable efficient and scalable generation (Esser et al., 2021; Rombach et al., 2022a; Chang et al., 2022; Yu et al., 2022b). By encoding images into discrete (van den Oord et al., 2017; Ryu, 2024) or continuous (Rombach et al., 2022a) latent tokens, these models avoid operating directly in pixel space and instead focus on learning semantically meaningful representations. Early work used autoencoders (Hinton & Salakhutdinov, 2006; Vincent et al., 2008) to learn low-dimensional latent representations, which were later extended to structured generative models such as VAEs and VQ-GAN (Van Den Oord et al., 2017; Razavi et al., 2019; Esser et al., 2021). VQ-GAN–style (Goodfellow et al., 2014; Esser et al., 2021; Yu et al., 2021; Zheng & Vedaldi, 2023; Yu et al., 2024a) discrete formulations naturally align with autoregressive (Esser et al., 2021) and masked generative models (Chang et al., 2022), facilitating the adoption of techniques originally developed for language modeling (Brown et al., 2020). Continuous tokenization follows the variational autoencoder (VAE) framework (Kingma & Welling, 2014a), in which latent representations are modeled as samples from a normal distribution.

**Image Generation.** Image generation methods are predominantly categorized into autoregressive and diffusion models. Early autoregressive approaches were primarily built upon convolutional neural networks (Van den Oord et al., 2016), and were later extended with Transformer-based architectures (Vaswani et al., 2017; Lee et al., 2022; Liu et al., 2024; Sun et al., 2024; Yu et al., 2025a) to improve scalability and modeling capacity (Chang et al., 2022; Tian et al., 2024). Diffusion models have demonstrated strong generative performance since their introduction (Sohl-Dickstein et al., 2015). Subsequent developments refined the denoising process and significantly improved sample quality (Nichol & Dhariwal, 2021; Dhariwal & Nichol, 2021a; Song et al., 2022). A pivotal advance in both performance and efficiency was achieved by

latent diffusion models (Vahdat et al., 2021; Rombach et al., 2022b), which leverage learned tokenizers to perform denoising in a compact latent space, thereby reducing computational cost while preserving visual fidelity (Van Den Oord et al., 2017; Esser et al., 2021; Peebles & Xie, 2023; Qiu et al., 2025). Recent research has further advanced image generation by improving tokenizer design (Chen et al., 2025; Zha et al., 2024; Yao & Wang, 2025) and by exploring hybrid frameworks that combine diffusion and autoregressive modeling paradigms (Li et al., 2024a).

## 3 Method

This section presents our method. We first review query-based 1D tokenization for latent image modeling in both discrete and continuous settings (section 3.1). We then introduce `SMAP`, a semantic-aware tokenizer that incorporates conditional semantics directly into token formation and reconstruction (section 3.2). Finally, we present `CARD`, a hybrid autoregressive–diffusion generator designed to exploit the information ordering induced by `SMAP` (section 3.3).

### 3.1 Preliminary: Query-Based 1D Tokenization

Recent token-based image representations increasingly adopt a query-based tokenization paradigm, drawing inspiration from `Q-Former`-style architectures (Li et al., 2023a; Yu et al., 2024b; Li et al., 2024b; Chen et al., 2025), in which a fixed set of learnable queries selectively attends to visual features to extract compact representations. Several recent visual tokenizers adopt this query-based formulation, among which `TiTok` is a representative example.

`TiTok` is a transformer-based, one-dimensional vector-quantized (`VQ`) tokenizer that departs from conventional grid-structured latent representations. Instead of preserving a two-dimensional spatial layout, `TiTok` represents an image using a compact sequence of latent tokens. Given an input image $\mathbf{I} \in \mathbb{R}^{H \times W \times 3}$, `TiTok` first applies a patch embedding operation with downsampling factor $f$, producing visual patch features $\mathbf{F} \in \mathbb{R}^{(\frac{H}{f} \times \frac{W}{f}) \times D}$. A set of learnable latent tokens $\mathbf{L} \in \mathbb{R}^{K \times D}$ is then concatenated with the patch tokens along the sequence dimension. The resulting sequence is processed by a Vision Transformer (`ViT`) encoder `Enc` to produce token embeddings, from which only the embeddings corresponding to the latent tokens are retained:

$$[\_; \mathbf{Z}_{1D}] = \text{Enc}([\mathbf{F}; \mathbf{L}]). \tag{1}$$

where $[\cdot; \cdot]$ denotes concatenation along the sequence dimension, $\mathbf{Z}_{1D} \in \mathbb{R}^{K \times D}$ represents the resulting one-dimensional latent tokens, and $\_$ denotes tokens that are discarded in subsequent processing.

The resulting one-dimensional latent tokens $\mathbf{Z}_{1D} \in \mathbb{R}^{K \times D}$ can be instantiated using discrete or continuous representations. In the original `TiTok` framework, latent tokens are quantized using a vector quantizer `Quant`$(\cdot)$, which maps each token to its nearest entry in a learnable codebook. Subsequent work (Kim et al., 2025) extends this formulation by modeling latent tokens as continuous random variables and applying variational regularization, producing a compact 1D VAE representation that avoids information loss induced by quantization. For notational convenience, we use a unified regularization operator `Regu`$(\cdot)$ to denote the latent regularization applied before decoding. Its concrete instantiation under the VQ, KL, and SoftVQ formulations is provided in Appendix B.

During de-tokenization, a sequence of mask tokens $\mathbf{M} \in \mathbb{R}^{(\frac{H}{f} \times \frac{W}{f}) \times D}$ is introduced and concatenated with the latent tokens, regardless of whether they are discrete or continuous. The combined sequence is then passed through a Vision Transformer decoder `Dec` to reconstruct the image $\hat{\mathbf{I}}$:

$$[\_; \hat{\mathbf{I}}] = \text{Dec}([\text{Regu}(\mathbf{Z}_{1D}); \mathbf{M}]). \tag{2}$$

### 3.2 SMAP: Semantic-Aware Prefix Tokenization with Semantics-Preserved Prefixes

**Overall Design.** SMAP is designed to force semantic information to become a functional prefix-level carrier of reconstruction, rather than an auxiliary alignment target. To this end, `SMAP` extends query-based

1D tokenization with two key ideas. First, semantic conditions are injected into both the encoder and decoder as explicit sequence elements, allowing semantic cues to participate in token formation and reconstruction. Second, a tail token dropping strategy is applied during training so that semantic conditions and early token prefixes must progressively absorb more global structural responsibility. Together, these two mechanisms encourage the tokenizer to learn information-ordered latent sequences with strong semantic grounding.

**Query-based Encoder–Decoder Formulation.** Given an input image $\mathbf{I}$, `SMAP` first extracts visual features $\mathbf{F}$ using a `ViT`-based image encoder. Similar to `TiTok`, a set of learnable latent queries $\{\mathbf{q}_i\}_{i=1}^{K}$ is used to aggregate visual information via self-attention, producing the output token sequence $\mathbf{z}_{1:K}^{[t]}$ from the $t$-th block. Similarly, in the decoder, information is propagated through self-attention between the learnable mask tokens $\{\mathbf{m}_i\}_{i=1}^{L}$ and the latent tokens $\{\hat{\mathbf{q}}_i\}_{i=1}^{K}$. Unlike `TiTok`, `SMAP` directly reconstructs the final image $\hat{\mathbf{I}}$ from the mask tokens $\{\mathbf{m}_i\}_{i=1}^{K}$ in the output of the decoder, rather than using latent tokens as primary reconstruction carriers. This design assigns latent tokens the role of encoding semantic and global information, while explicitly delegating spatially structured image synthesis to the mask tokens.

Moreover, `SMAP` supports both discrete and continuous forms of token regularization within a unified tokenizer framework, enabling flexible training across different generative paradigms such as autoregressive and diffusion models. By jointly designing the query-based tokenization mechanism and the ViT-based encoder–decoder architecture, `SMAP` can more fully exploit the scaling properties of `Transformer` models. Figure 3(a) illustrates the scaling behavior of `SMAP`, while Figure 3(b) compares the current design against prior tokenizers. In contrast to previous approaches (Yu et al., 2024b; Miwa et al., 2025) that rely on multi-stage optimization or additional pretraining procedures, our tokenizer significantly simplifies the training pipeline in a single stage while maintaining strong reconstruction quality and scalability to large datasets and model sizes.

**Semantic Injection Mechanism.** Although methods such as `REPA` encourage tokenizer representations to correlate with semantic features to some extent, they typically treat semantic signals as auxiliary alignment or regularization objectives, without explicitly requiring semantics to bear essential informational responsibility during reconstruction and representation learning. To make semantic information an indispensable component of tokenizer pre-training, we introduce an explicit semantic injection mechanism. We first discuss the construction of conditional embeddings $\mathbf{C} \in \mathbb{R}^{N \times D}$ from semantic supervision. For class-level conditions, we jointly train an additional class embedding module within the tokenizer, whose embedding dimensionality is aligned with that of the learnable latent tokens, allowing direct concatenation and interaction along the sequence dimension.

As shown in Figure 2(a), we derive conditional embeddings $\mathbf{C} \in \mathbb{R}^{(N \times D)}$ from class labels and insert them between visual patch tokens $\mathbf{V} \in \mathbb{R}^{(L \times D)}$ and learnable latent queries $\mathbf{L} \in \mathbb{R}^{(K \times D)}$. The resulting token sequence is then jointly processed by the encoder `Enc`, allowing visual content and explicit semantic cues to interact through self-attention and jointly shape the formation of latent token representations.

$$[\_ \, ; \_ \, ; \mathbf{Z}_{1\mathrm{D}}] = \mathtt{Enc}([\mathbf{V}; \mathbf{C}; \mathbf{L}]). \tag{3}$$

In the de-tokenization stage, `SMAP` symmetrically incorporates the semantic embeddings introduced during tokenization. As illustrated in the de-tokenization module in Figure 2(a), the conditional embeddings are injected between the learnable mask tokens $\mathbf{M} \in \mathbb{R}^{L \times D}$ and the processed latent tokens $\hat{\mathbf{L}} \in \mathbb{R}^{K \times D}$, and jointly modeled through the decoder `Dec`, enabling the mask tokens to aggregate the information required for reconstruction and ultimately generate the image $\hat{\mathbf{I}}$.

$$[\hat{\mathbf{I}}; \_ \, ; \_ \, ;] = \mathtt{Dec}([\mathbf{M}; \mathbf{C}; \hat{\mathbf{L}}]). \tag{4}$$

**Tail token dropping.** To enforce semantic dependency during tokenizer pre-training, we introduce a *tail token dropping* strategy that perturbs the latent token sequence at training time. Let the encoder output latent tokens be $\mathbf{Z}_{1:K}^{\mathtt{1D}}$. At each iteration, we sample a retained prefix length $k \in \{0, 1, \ldots, K\}$ and keep only the prefix tokens $\mathbf{Z}_{1:k}^{\mathtt{1D}}$, while removing the tail tokens $\mathbf{Z}_{k+1:K}^{\mathtt{1D}}$ (or equivalently masking them out in the attention computation). The extreme case $k = 0$ corresponds to dropping all latent tokens, in which case the decoder must reconstruct the image $\hat{\mathbf{I}}$ using only the conditional embeddings $\mathbf{C} \in \mathbb{R}^{N \times D}$ together with

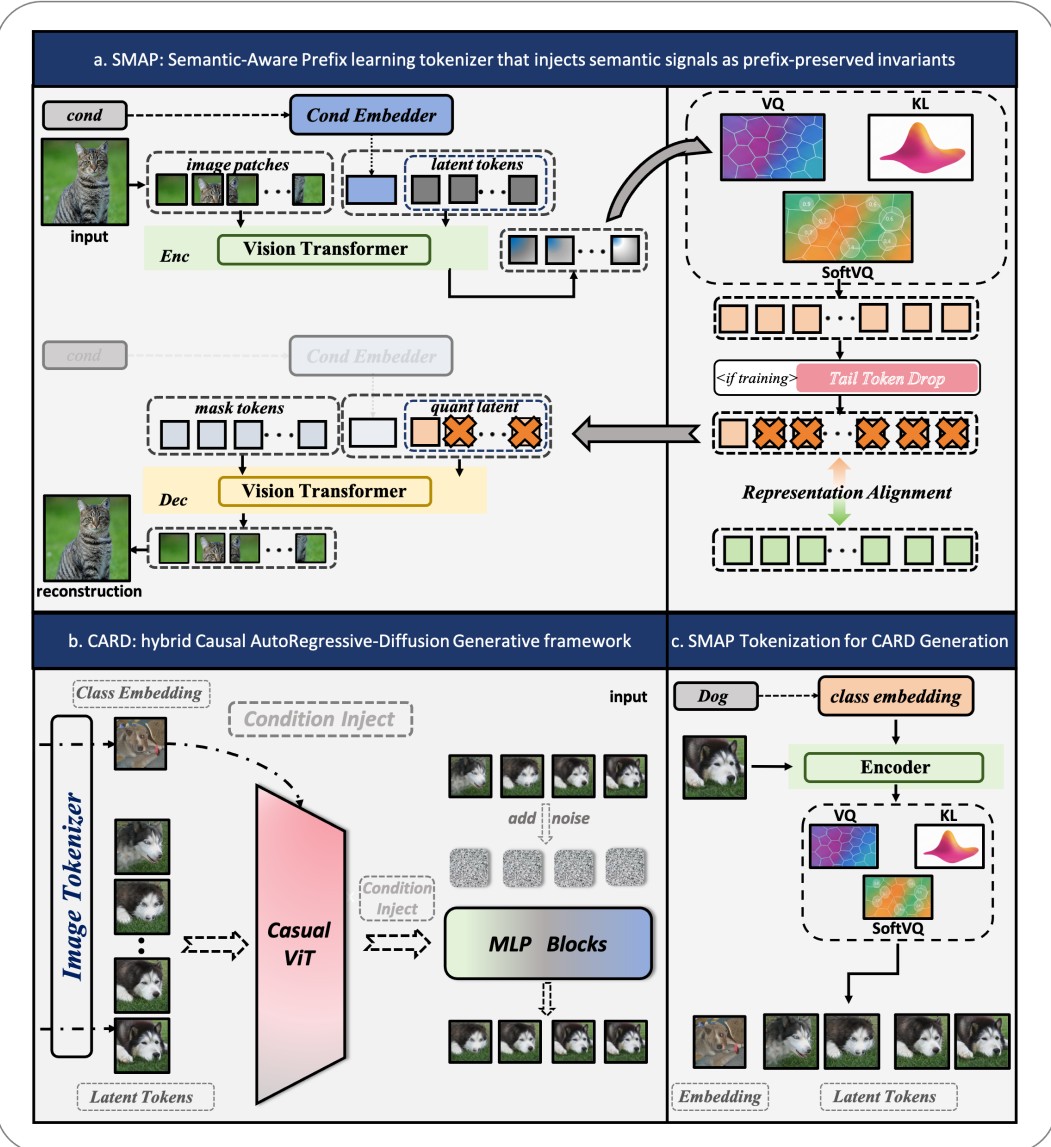

Figure 2: Overview of our method. **(a)** proposes a novel mechanism for semantic injection. It extracts conditional embeddings from class labels and inserts them between visual patch tokens and learnable latent tokens. The condition embeddings act as intermediaries that interact jointly with image patches to guide the formation of latent tokens. It further strengthens semantic dependency through a tail token dropping strategy. **(b)** proposes a hybrid Causal AutoRegressive–Diffusion framework that fully leverages `SMAP`'s capabilities. **(c)** shows the `SMAP` tokenization process for `CARD` generation.

the mask tokens $\mathbf{M} \in \mathbb{R}^{K \times D}$. This training-time perturbation explicitly increases the informational burden placed on the conditional embeddings. Importantly, this strategy operates *directly on the token sequence*, so we can construct the decoder input by concatenating the retained latent prefix $\texttt{Regu}(\mathbf{Z}_{1:k}^{\text{1D}})$ with the semantic embeddings $\mathbf{C}$ and the learnable mask tokens $\mathbf{M}$.

$$[\hat{\mathbf{I}}; \_\_ ; \_\_ ;] = \texttt{Dec}([\mathbf{M}; \mathbf{C}; \texttt{Regu}(\mathbf{Z}_{1:k})]) \tag{5}$$

The prefix length $k$ is sampled only during training. Specifically, we draw $k$ from a uniform distribution over token indices $k \sim \texttt{Unif}\{0, 1, \ldots, K\}$, so that different token budgets are randomly explored across training iterations. Consequently, the semantic prefix becomes the only information pathway that is preserved across all sampled token budgets.

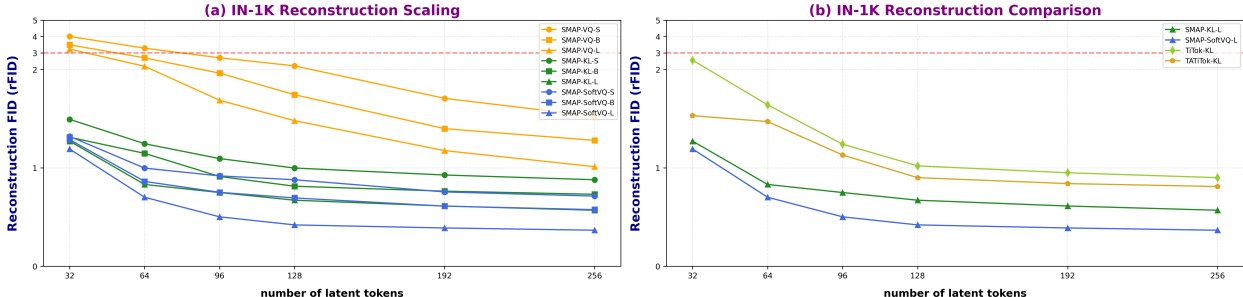

Figure 3: **ImageNet-1K reconstruction scaling and comparison. (a)** Reconstruction FID (rFID) of `SMAP` under different token budgets and model scales. Across VQ, KL, and SoftVQ variants, increasing the number of latent tokens consistently improves reconstruction quality, and larger `SMAP` models achieve stronger performance under the same token budget. **(b)** Reconstruction comparison with prior 1D tokenizers. At matched token lengths, `SMAP` consistently outperforms `TiTok` and `TA-TiTok`, with the largest gains observed in continuous latent settings. Overall, the results show that `SMAP` scales favorably with both token budget and model capacity, while providing substantially better reconstruction quality than existing baselines.

### 3.3 CARD: Hybrid Diffusion–Autoregressive Generative Model

**Architecture.** To fully exploit the semantic-aware and information-ordered latent space learned by `SMAP`, we propose `CARD`, a hybrid generative framework that combines causal autoregressive modeling with diffusion-style refinement. As illustrated in Figure 2(b), `CARD` first applies a causal transformer to model the structural dependencies among latent tokens in an autoregressive manner, thereby capturing coarse global structure and long-range token interactions. The autoregressive predictions are then passed to a lightweight continuous refinement module, instantiated as a stack of `MLP` blocks, which denoises noisy latent variables and improves generation fidelity.

Concretely, let $\mathbf{Z}_{1D}$ denote the latent token sequence. The causal autoregressive module produces structure-aware latent predictions $\mathtt{AR}(\mathbf{Z}_{1D})$, which are used as conditional inputs to the refinement model. Given a noisy latent $\mathbf{x}_t$ at timestep $t$, the denoising velocity is predicted as

$$\mathbf{v}_t = \mathtt{MLP}\big(\mathbf{x}_t,\, t,\, \mathtt{AR}(\mathbf{Z}_{1D})\big), \tag{6}$$

where $\mathbf{x}_t$ is the noisy latent variable and $\mathtt{AR}(\cdot)$ denotes the autoregressive outputs. In contrast to `MAR` (Li et al., 2024a), which directly concatenates condition embeddings with image tokens, `CARD` injects conditions into the generator through adaptive normalization, following the conditioning strategy of `DiT` (Peebles & Xie, 2023). This design preserves the compact token structure while enabling flexible conditional control.

**Semantic Condition Sharing.** A key design choice of `CARD` is that its conditioning signal is not introduced through a separately trained class encoder. Instead, as illustrated in Figure 2(c), we directly reuse the class-aware semantic embedding learned during `SMAP` pretraining as the condition input for generation. More specifically, the class label is first mapped to a semantic embedding by the same condition embedding module used in the tokenizer, and this embedding is then paired with the latent tokens produced by `SMAP`. The resulting shared semantic space is subsequently used throughout `CARD`, ensuring that the condition signal used in generation is consistent with the semantic prefix that shaped tokenizer learning.

This semantic condition sharing has two advantages. First, it removes the need to introduce an additional condition encoder on the generator side, thereby simplifying the overall architecture. Second, it strengthens semantic consistency between tokenization and generation: the same embedding space that guides semantic prefix learning in `SMAP` is also used to control downstream image synthesis in `CARD`. Empirically, this sharing mechanism improves the alignment between class conditions and generated content, and further demonstrates that the semantic representations learned by `SMAP` are transferable and functionally useful for downstream generation. Detailed empirical analysis of this design is provided in Section 4.

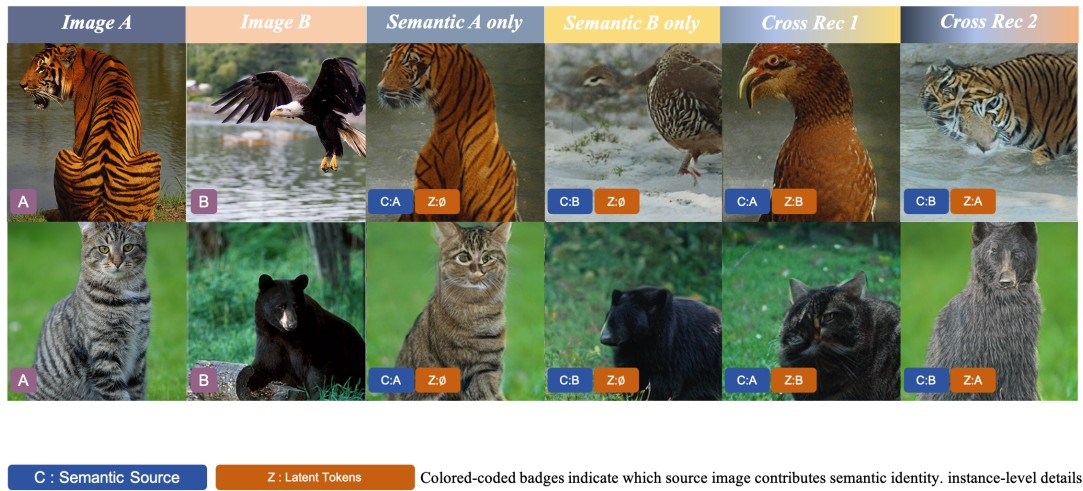

Figure 4: **Semantic identity is controlled by $C$, while instance-level details are carried by $Z$.** We visualize reconstructions obtained by independently manipulating the semantic condition $C$ and latent tokens $Z$. Using only $C$ with $Z = \emptyset$ yields coarse reconstructions that preserve category-level semantics. In contrast, cross-combining $C$ from one image with $Z$ from another transfers semantic identity and instance-specific appearance in a complementary manner.

## 4 Experiments

### 4.1 Experiments Setup

**Implementation Details of Tokenizer.** We use `SoftVQ` codebase (Chen et al., 2025) to train `SMAP`. We instantiate three variants of `SMAP`, all sharing the same encoder–decoder architecture but differing in model scale, with parameter counts of 185M, 391M, and 568M, corresponding to the `SMAP-S`, `SMAP-B`, and SMAP-L configurations, respectively. We consider three latent regularization schemes: `VQ` (van den Oord et al., 2017; Yu et al., 2022a), `SoftVQ` (Chen et al., 2025), and `KL` (Takahashi et al., 2019). For the `VQ` variant, we adopt a codebook size of 8192 with a channel dimension of 64, and train models with latent token lengths of 64 and 128 to align with the settings used in `TiTok`. For the `KL`-based variant, following the design of `MAR` (Li et al., 2024a), we model latent representations as continuous features with 16 channels, and consider latent token lengths of 128 and 256. For the `SoftVQ` variant, we employ a hierarchical codebook design (Li et al., 2024b) with four levels and a total codebook size of 8192, while keeping the channel dimension consistent with the KL variant (*i.e.*, 16 channels). For ablation studies, we additionally evaluate smaller token budgets (32, 64, 96 and 128) to analyze the effect of compact latent representations. For the main comparison and generator experiments, we use 128 tokens unless otherwise specified. Please refer to Appendix A for additional experimental details.

**Implementation Details of Generator.** For the discrete variants of `SMAP`, we adopt `LlamaGen` (Sun et al., 2024) as the generative model to evaluate generation performance, following standard practice for discrete token-based representations. For the continuous variants of `SMAP`, we employ our proposed `CARD` as the generator, which is specifically designed to match the inductive bias and information ordering induced by `SMAP`. We consider three variants of `CARD`, namely `CARD-B`, `CARD-L`, and `CARD-XL`, with 234M, 568M, and 1.1B parameters, respectively. Detailed architectural configurations for each variant are provided in Table 1.

**Evaluation Metrics.** Our evaluation protocol closely follows prior work (Yu et al., 2022b). For the reconstruction evaluation of `SMAP`, we report reconstruction Frechet Inception Distance (FID) (Heusel et al., 2017) and Inception Score (IS) (Salimans et al., 2016) on the ImageNet (Deng et al., 2009) validation set,

Table 1: **Architecture Configuration of CARD.** Following `MAR`, we scale up blocks across three configurations.

| Model | Depth | Width | Heads | $D_{\mathrm{mlp}}$ | $W_{\mathrm{mlp}}$ | #Params |
|---|---|---|---|---|---|---|
| CARD-B | 24 | 768 | 12 | 6 | 1024 | 234M |
| CARD-L | 32 | 1024 | 16 | 8 | 1280 | 568M |
| CARD-XL | 40 | 1280 | 16 | 12 | 1536 | 1.1B |

Table 2: **Ablation on the improved one-stage training recipe.** All models are trained and evaluated on `ImageNet256`. We compare the baseline tokenizer (`TiTok`) against `SMAP` under matched token budgets. Numbers in parentheses indicate the change relative to the baseline. Lower rFID and higher IS are better.

| arch | tokens # | c | **TiTok** rFID ↓ | IS↑ | **SMAP** rFID ↓ | IS↑ |
|---|---|---|---|---|---|---|
| VQ | 32 | – | 7.72 | 98.3 | 3.24 (-4.48) | 230.4 (+132.1) |
| | 64 | – | 4.25 | 138.0 | 2.29 (-1.96) | 260.1 (+122.1) |
| | 128 | – | 2.63 | 168.1 | 1.47 (-1.16) | 280.5 (+112.4) |
| KL | 32 | 16 | 2.56 | 171.7 | 1.07 (-1.49) | 280.4 (+108.7) |
| | 64 | 16 | 1.64 | 198.0 | 0.96 (-0.68) | 299.5 (+101.5) |
| | 128 | 16 | 1.02 | 209.7 | 0.75 (-0.27) | 308.9 (+99.2) |

providing a comprehensive evaluation of reconstruction fidelity and perceptual quality. To avoid ambiguity, we explicitly distinguish generation and reconstruction FID throughout the paper, denoted as gFID and rFID, respectively. To evaluate generative performance, we train `CARD` on the latent representations produced by each variant of `SMAP`. We report gFID and IS computed over 50,000 generated samples, following the evaluation protocol of `ADM` (Dhariwal & Nichol, 2021b). Detailed experimental results are provided in Appendix C.

### 4.2 Optimized Image Tokenization with SMAP

**Improved One-Stage Training Recipe.** Table 2 summarizes the performance gains of our improved one-stage training recipe over the original schemes in (Yu et al., 2024b). We observe that the proposed one-stage training consistently outperforms the original `TiTok` in both the `VQ` and `KL` variants, yielding a uniformly lower rFID in all evaluated token lengths. This shows that the improvement is not tied to a specific latent formulation or token budget, but reflects a more robust and effective tokenizer training strategy.

**Semantic Understanding through Conditional Embedding Injection.** We perform a multi-stage analysis to verify that conditional embedding injection is not merely incidental, but instead plays a functional role in both tokenizer learning and downstream generation.

We first examine whether the injected semantic condition can itself serve as a meaningful source of global structure. As shown in Figure 1, when only the class condition is provided to the decoder, `SMAP` is already able to reconstruct coarse images that capture recognizable category-level semantics and rough global layout. Although these reconstructions are still blurry and lack instance-specific details, they are far from arbitrary outputs: the reconstructed content already reflects the semantic commonalities associated with the conditioning signal. This indicates that the semantic embedding is not treated merely as side information, but is explicitly trained to carry information that is directly useful for reconstruction. Besides, we study how semantic information interacts with latent tokens once additional latent capacity is introduced. Again in Figure 1, adding latent tokens on top of the class condition leads to consistent improvements in reconstruction fidelity. This behavior reveals a clear division of roles: the conditional embedding establishes category-level identity and coarse global structure, while latent tokens progressively recover finer instance-level appearance, texture, and spatial details. Importantly, the semantic signal remains effective throughout this process

Table 3: **Ablation on progressive token truncation.** All models are trained and evaluated on `ImageNet256`. We compare `SMAP` trained without and with progressive token truncation. Numbers in parentheses indicate the change relative to the corresponding model trained without truncation. Lower rFID and higher IS are better.

| arch | tokens # | c | SMAP w/o trunc. rFID↓ | IS↑ | SMAP w/ trunc. rFID↓ | IS↑ |
|------|----------|---|-----------------------|------|----------------------|------|
| VQ | 32 | – | 3.41 | 220.8 | 3.24 (-0.17) | 230.4 (+9.6) |
| | 64 | – | 2.50 | 253.4 | 2.29 (-0.21) | 260.1 (+6.7) |
| | 128 | – | 1.65 | 269.8 | 1.47 (-0.18) | 280.5 (+10.7) |
| KL | 32 | 16 | 1.20 | 269.1 | 1.07 (-0.13) | 280.4 (+11.3) |
| | 64 | 16 | 1.03 | 278.6 | 0.96 (-0.07) | 299.5 (+20.9) |
| | 128 | 16 | 0.75 | 298.4 | 0.69 (-0.06) | 308.9 (+10.5) |

Table 4: **System-level comparison on ImageNet 256×256 conditional generation.** `SMAP+CARD` achieves competitive performance under a compact 128-token budget across both KL and SoftVQ variants. "Model (G)" denotes the generator, "# Params (G)" its parameter count, "Model (T)" the tokenizer, "# Params (T)" its parameter count, and "# Tokens" the number of latent tokens used during generation. [†] indicates that the model was trained on data beyond ImageNet.

| Model (G) | # Params (G) | Model (T) | # Params (T) | # Tokens↓ | rFID↓ | w/o CFG gFID↓ | IS↑ | w/ CFG gFID↓ | IS↑ |
|-----------|--------------|-----------|--------------|-----------|-------|---------------|-----|--------------|-----|
| *Auto-regressive* | | | | | | | | | |
| VQGAN (Esser et al., 2021) | 1.4B | VQ | 23M | 256 | 7.94 | – | – | 5.20 | 290.3 |
| ViT-VQGAN (Yu et al., 2021) | 1.7B | VQ | 64M | 1024 | 1.28 | 4.17 | 175.1 | – | – |
| LlamaGen-3B (Sun et al., 2024) | 3.1B | VQ | 72M | 576 | 2.19 | – | – | 2.18 | 263.3 |
| TiTok-S-128 (Yu et al., 2024b) | 287M | VQ | 72M | 128 | 1.61 | – | – | 1.97 | 281.8 |
| VAR (Tian et al., 2024) | 2B | MSRQ[†] | 109M | 680 | 0.90 | – | – | 1.92 | 323.1 |
| MAR-H (Li et al., 2024a) | 943M | KL | 66M | 256 | 1.22 | 2.35 | 227.8 | 1.55 | 303.7 |
| *Diffusion-based* | | | | | | | | | |
| LDM-4 (Vahdat et al., 2021) | 400M | KL[†] | 55M | 4096 | 0.27 | 10.56 | 103.5 | 3.60 | 247.7 |
| MDTv2-XL/2 (Sahoo et al., 2024) | 676M | – | – | – | – | 5.06 | 155.6 | 1.58 | 314.7 |
| DiT-XL/2 (Peebles & Xie, 2023) | 675M | – | – | – | – | 9.62 | 121.5 | 2.27 | 278.2 |
| SiT-XL/2 (Ma et al., 2024) | 675M | – | – | – | – | 8.30 | 131.7 | 2.06 | 270.3 |
| + REPA (Yao et al., 2024) | 675M | – | – | – | – | 5.90 | 157.8 | 1.42 | 305.7 |
| TexTok-256 (Zha et al., 2024) | 675M | KL | 176M | 256 | 0.69 | – | – | 1.46 | 303.1 |
| LightningDiT (Yao & Wang, 2025) | 675M | KL | 70M | 256 | 0.28 | 2.17 | 205.6 | 1.35 | 295.3 |
| MAETok + LightningDiT | 675M | AE | 176M | 128 | 0.48 | 2.21 | 208.3 | 1.73 | 308.4 |
| MAETok + SiT-XL | 675M | AE | 176M | 128 | 0.48 | 2.31 | 216.5 | 1.67 | 311.2 |
| *Ours* | | | | | | | | | |
| SMAP(VQ) + LlamaGen (Sun et al., 2024) | 3.1B | VQ | 185M | 128 | 1.47 | 2.86 | 233.7 | 2.14 | 290.5 |
| SMAP(KL) + CARD | 568M | KL | 391M | 128 | 0.75 | 2.38 | 244.6 | 1.97 | 320.8 |
| SMAP(KL) + CARD | 1.1B | KL | 391M | 128 | 0.75 | 2.34 | 251.4 | 1.85 | 325.1 |
| SMAP(SoftVQ) + CARD | 568M | SoftVQ | 391M | 128 | 0.55 | 2.69 | 211.3 | 2.01 | 304.8 |
| SMAP(SoftVQ) + CARD | 1.1B | SoftVQ | 391M | 128 | 0.55 | 2.28 | 245.3 | 1.79 | 328.9 |

rather than being overridden as more latent tokens are introduced, suggesting that semantic information is preserved as a stable prefix-level component of the learned representation.

To further probe this role decomposition, we visualize cross reconstructions in Figure 4, where the semantic source $C$ and latent tokens $Z$ are independently manipulated across two input images. When only the semantic condition is retained ($Z = \emptyset$), the model still produces semantically recognizable reconstructions, again confirming that semantic identity can be recovered from the learned conditional prefix alone. When $C$ from one image is combined with $Z$ from another, the resulting reconstruction follows the semantic identity specified by $C$ while inheriting instance-level appearance cues from $Z$. In other words, the semantic condition primarily determines category-level identity, whereas latent tokens contribute instance-specific visual details. This provides direct evidence that `SMAP` learns a semantically grounded representation in which semantic prefixes and latent tokens play complementary and clearly differentiated roles.

Taken together, these findings show that conditional embedding injection does more than provide weak semantic alignment. Instead, it realizes semantic-aware prefix learning: semantic conditions are forced to encode category-level identity and global structure, while latent tokens progressively refine the representation with instance-level detail, yielding a latent space that benefits reconstruction.

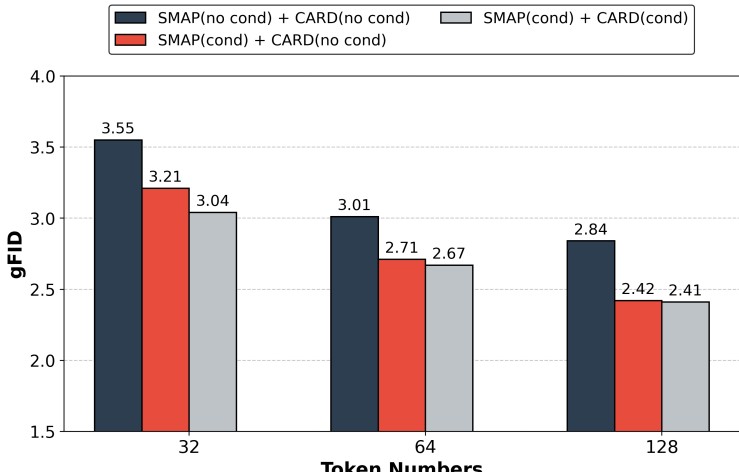

Figure 5: **Effect of semantic-aware tokenization on downstream generation.** We compare three settings: a reconstruction-only tokenizer with independent generator conditioning, a semantic-aware tokenizer with independent generator conditioning, and a shared-semantic setting in which the generator reuses the tokenizer's learned semantic embedding space. Semantic-aware tokenizer pretraining consistently improves gFID across all token budgets, and semantic sharing yields a further gain in every setting.

**Enforcing Semantic Dependency with Progressive Token Truncation.** The emergence of semantically meaningful representations in `SMAP` is not solely a consequence of architectural design, but is critically driven by the proposed progressive token truncation strategy. By truncating the suffix of the latent token sequence during training, the model is forced to shift increasing reconstruction responsibility toward the semantic condition and the early latent prefix. As a result, the decoder can no longer rely exclusively on the full latent token set, and must instead learn to recover category-level identity and coarse global structure from the conditional prefix itself. This behavior is clearly illustrated in Figure 1. When only the class condition is provided, the model already produces coarse yet semantically recognizable reconstructions, indicating that the semantic prefix has absorbed meaningful global structural information. When latent tokens are added back, reconstruction fidelity improves substantially and instance-specific details progressively reappear. This shows that progressive token truncation does not simply act as a generic regularizer; rather, it explicitly encourages an information-ordered representation in which semantic conditions provide the structural scaffold and latent tokens refine it with finer visual detail.

The quantitative results in Table 3 support the same conclusion. Across all VQ settings, progressive token truncation improves both rFID and IS, with especially clear gains under limited token budgets. For example, under VQ tokenization, truncation reduces rFID from 3.41 to 3.24 at 32 tokens and from 2.50 to 2.29 at 64 tokens, while increasing IS at every token budget. A similar trend is observed for KL tokenization at 32 and 64 tokens, where both rFID and IS improve with truncation. At the 128-token setting, both VQ and KL tokenizers already achieve strong performance within their respective formulations, indicating that this regime is not constrained by limited latent capacity.

Taken together, these qualitative and quantitative results show that progressive token truncation is the key mechanism that makes semantic information indispensable during training. Without it, semantic embeddings are much less likely to develop into a functional reconstruction pathway; with it, they are explicitly forced

to encode category-level commonality and global structure, thereby realizing semantic-aware prefix learning rather than merely adding auxiliary semantic supervision.

### 4.3 Effect of Semantic-Aware Tokenization on Generation

**Semantic-aware tokenization improves downstream generation.** We next examine whether the semantic structure learned during tokenizer pretraining carries over to downstream generation. To isolate this effect, we train `CARD` on latent sequences produced by `SMAP` under three settings: (i) a reconstruction-only tokenizer with an independent generator-side conditioning pathway, (ii) a semantic-aware tokenizer while keeping the generator conditioning independent, and (iii) a shared-semantic setting in which the generator reuses the semantic embedding space learned during tokenizer pretraining.

As shown in Figure 5, the benefit of semantic-aware tokenization is already evident even without semantic sharing on the generator side. Replacing the reconstruction-only tokenizer with the semantic-aware variant consistently improves gFID across all token budgets, from 3.55 to 3.21 at 32 tokens, from 3.01 to 2.71 at 64 tokens, and from 2.84 to 2.42 at 128 tokens. This trend suggests that the gain is not solely due to a stronger conditioning mechanism during generation. Instead, semantic-aware tokenizer pretraining itself yields a latent space that is easier for the generator to model. Reusing the tokenizer's learned semantic embedding space in the generator brings a further, though smaller, improvement. With semantic sharing, gFID is further reduced from 3.21 to 3.04 at 32 tokens, from 2.71 to 2.67 at 64 tokens, and from 2.42 to 2.41 at 128 tokens. While these additional gains are more modest than those obtained from semantic-aware tokenization itself, their consistency indicates that the semantic representation learned by `SMAP` is not only transferable, but also directly useful for downstream generation.

This effect is also reflected at the system level in Table 4. Under a compact 128-token budget, `SMAP+CARD` achieves competitive conditional generation performance against strong autoregressive and diffusion-based baselines. In particular, the 1.1B `SMAP(KL)` + `CARD` model reaches gFID = 1.85 with CFG, while the 1.1B `SMAP(SoftVQ)` + `CARD` model further improves to gFID = 1.79 and IS = 328.9. These results are obtained with substantially fewer latent tokens than many prior methods, including approaches based on 256, 576, 680, 1024, or 4096 tokens, suggesting that the semantic-aware latent structure learned by `SMAP` supports efficient and high-quality generation.

Taken together, these results suggest two complementary advantages of semantic-aware prefix learning. First, it produces a more generator-friendly latent representation even when tokenization and generation are trained separately. Second, the learned semantic space can be reused by the generator to further improve alignment between tokenizer pretraining and downstream synthesis. Overall, the benefit of semantic-aware tokenization is therefore not limited to reconstruction quality; it also leads to a latent representation that transfers more effectively to generative modeling.

## 5 Conclusion

We presented `SMAP`, a semantic-aware tokenizer that makes semantic information a functional component of tokenizer pretraining rather than a weak alignment signal. Through conditional embedding injection and tail token dropping, `SMAP` learns semantically grounded, information-ordered latent representations that improve reconstruction quality across discrete and continuous tokenization settings. Building on this latent space, we further introduced `CARD`, a hybrid autoregressive–diffusion generator that leverages the learned semantic structure for conditional image synthesis. Experiments on ImageNet show that semantic-aware prefix learning benefits both tokenizer reconstruction and downstream generation, suggesting that semantics should be treated as an integral part of representation learning in latent image modeling.

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

## Appendix

In the supplementary materials, we provide the following additional details:

- The comprehensive training and testing hyper-parameters and training costs for `SMAP`(section A).

- The detailed instantiation of the unified regularization operator `Regu`(·) under both discrete and continuous tokenization settings (section B).

- A more comprehensive comparison with more metrics and baselines (section C).

- Limitation discussion (section D).

- Dataset Licenses (section E).

## A  Additional Implementation Details

**Tokenizer training.** For image reconstruction (**tokenizer**), we train `SMAP` on ImageNet following the one-stage recipe described in the main paper. Unless otherwise specified, the training augmentation is limited to random resized cropping and horizontal flipping. All tokenizer variants are trained for 500K iterations at resolutions $256 \times 256$ and $512 \times 512$, respectively. We use the AdamW optimizer Loshchilov & Hutter (2017) with batch size `256`, initial learning rate $1 \times 10^{-4}$, and weight decay $1 \times 10^{-4}$. The learning rate follows a cosine decay schedule with `20` warm-up epochs. We use patch size 16 for all Vision Transformer tokenizers at resolution $256 \times 256$, and increase it to 32 at resolution $512 \times 512$ for better computational efficiency.

`SMAP`-S, `SMAP`-B, and `SMAP`-L denote the small, base, and large tokenizer variants, with parameter counts of 185M, 391M, and 568M, respectively. For the `VQ` variant, we use a codebook of size 8192 with code dimension 64, and train models with token lengths 64 and 128. For the `KL` variant, following Li et al. (2024a), we use continuous latent features with 16 channels and token lengths 128 and 256. For the `SoftVQ` variant, we adopt a four-level hierarchical codebook with total size 8192, while keeping the channel dimension at 16; the token lengths are also set to 128 and 256. During training, the retained prefix length $k$ is sampled uniformly from $\{0, 1, \ldots, K\}$, such that the decoder is exposed to variable token budgets throughout optimization. The class embedding module is trained jointly with the tokenizer, and the same semantic condition embedding is used in both the encoder and decoder.

**Generator training.** For image generation (**generator**), we use different generators for discrete and continuous latent spaces. For discrete latent tokenization, we adopt `LlamaGen` Sun et al. (2024) following the standard setup for class-conditional generation. For continuous latent tokenization, we train the proposed `CARD` model on top of the latent space produced by `SMAP`. `CARD`-B, `CARD`-L, and `CARD`-XL contain 234M, 568M, and 1.1B parameters, respectively, and their detailed architectural configurations are reported in Table 1. Unless otherwise specified, the generator is trained with batch size `2048` for `250k` iterations using AdamW Loshchilov & Hutter (2017), with learning rate $2 \times 10^{-4}$ and weight decay $1 \times 10^{-5}$. We use cosine learning-rate decay and apply class-condition dropout with probability `0.1` for classifier-free guidance.

For `CARD`, the class condition is not produced by a separate encoder. Instead, we directly reuse the semantic embedding module learned during `SMAP` pretraining, which ensures semantic consistency between tokenization and generation. During evaluation, we follow prior work Dhariwal & Nichol (2021b); Esser et al. (2021) and report gFID and IS over 50,000 generated samples. For class-conditional sampling, we use classifier-free guidance with guidance scale `[2.7 for 256]` at $256 \times 256$ and `[3.5 for 512]` at $512 \times 512$. For the discrete `LlamaGen` results, we use the decoding and sampling hyper-parameters from the official implementation unless otherwise specified. For the continuous `CARD` results, we use `25` flow-matching sampling steps as the default inference configuration.

**Training cost.** The tokenizer training takes 32 A800 GPUs for `32` hours for `SMAP`-S, 32 A800 GPUs for `32` hours for `SMAP`-B, and 32 A800 GPUs for `72` hours for `SMAP`-L. The generator training takes 16 A800 GPUs for `48` hours for `CARD`-B, 32 A800 GPUs for `48` hours for `CARD`-L, and 32 A800 GPUs for `96` hours for `CARD`-XL.

Table 5: **Instantiation of** Regu$(\cdot)$ **under different tokenizer formulations.**

| Tokenizer | Encoder output | Regularization target | Regu$(\cdot)$ instantiation | Output to decoder |
|---|---|---|---|---|
| VQ | $\mathbf{Z}_{1D} \in \mathbb{R}^{K \times D}$ | discrete codebook | nearest-neighbor vector quantization | quantized code embeddings |
| KL | $(\boldsymbol{\mu}, \boldsymbol{\sigma})$ | Gaussian posterior | reparameterized latent sampling | continuous sampled latents |
| SoftVQ | $\mathbf{Z}_{1D} \in \mathbb{R}^{K \times D}$ | soft code assignment | temperature-controlled soft quantization | weighted codebook mixtures |

## B Instantiation of the Unified Regularization Operator Regu$(\cdot)$

In the main text, we use Regu$(\cdot)$ as a unified notation for the latent regularization applied to the 1D latent tokens before decoding. This abstraction allows us to describe discrete and continuous tokenizers within a single encoder–decoder formulation. In practice, however, as shown in Table 5, Regu$(\cdot)$ corresponds to different operations depending on the tokenizer instantiation.

Let $\mathbf{Z}_{1D} = [\mathbf{z}_1, \ldots, \mathbf{z}_K] \in \mathbb{R}^{K \times D}$ denote the latent sequence produced by the encoder.

**VQ instantiation.** For the discrete VQ tokenizer, Regu$(\cdot)$ denotes vector quantization with a learned codebook $\mathcal{E} = \{\mathbf{e}_1, \ldots, \mathbf{e}_{|\mathcal{E}|}\}$. Each latent token $\mathbf{z}_i$ is replaced by its nearest codebook entry:

$$\text{Regu}(\mathbf{z}_i) = \mathbf{e}_{j^\star}, \qquad j^\star = \arg\min_j \|\mathbf{z}_i - \mathbf{e}_j\|_2^2. \tag{7}$$

Applying this token-wise operation to the full sequence yields the quantized latent sequence

$$\text{Regu}(\mathbf{Z}_{1D}) = [\mathbf{e}_{j_1^\star}, \ldots, \mathbf{e}_{j_K^\star}]. \tag{8}$$

This is the discrete latent representation used by the decoder.

**KL instantiation.** For the continuous VAE-style tokenizer, Regu$(\cdot)$ denotes variational regularization via Gaussian reparameterization. The encoder predicts mean and variance parameters $(\boldsymbol{\mu}_i, \boldsymbol{\sigma}_i)$ for each latent token, and sampling is performed as

$$\text{Regu}(\mathbf{z}_i) = \boldsymbol{\mu}_i + \boldsymbol{\sigma}_i \odot \boldsymbol{\epsilon}_i, \qquad \boldsymbol{\epsilon}_i \sim \mathcal{N}(\mathbf{0}, \mathbf{I}). \tag{9}$$

Thus, for the KL case, Regu$(\mathbf{Z}_{1D})$ denotes the reparameterized continuous latent sequence passed to the decoder. During training, this is accompanied by the standard KL regularization term that encourages the posterior to remain close to a Gaussian prior.

**SoftVQ instantiation.** For the SoftVQ tokenizer, Regu$(\cdot)$ denotes differentiable soft quantization rather than hard nearest-neighbor assignment. Each latent token $\mathbf{z}_i$ is softly matched against the codebook entries, producing assignment weights

$$\alpha_{ij} = \frac{\exp(-d(\mathbf{z}_i, \mathbf{e}_j)/\tau)}{\sum_{j'} \exp(-d(\mathbf{z}_i, \mathbf{e}_{j'})/\tau)}, \tag{10}$$

where $d(\cdot, \cdot)$ is a distance function and $\tau$ is a temperature parameter. The regularized latent is then given by the weighted combination

$$\text{Regu}(\mathbf{z}_i) = \sum_j \alpha_{ij} \mathbf{e}_j. \tag{11}$$

Accordingly, Regu$(\mathbf{Z}_{1D})$ is a differentiably quantized latent sequence that retains codebook structure while remaining continuous during optimization.

**Unifying view.** Although these three instantiations differ algorithmically, they play the same functional role in our framework: they transform the encoder-produced latent sequence into the regularized representation consumed by the decoder. Using Regu$(\cdot)$ therefore lets us present the SMAP tokenizer in a unified way while preserving the flexibility to instantiate it with discrete, continuous, or softly quantized latents.

Table 6: **Detailed results of preliminary experiments in the main paper.**

(a) reconstruction FID (`VQ`).

| #token | 32 | 64 | 96 | 128 | 192 | 256 |
|---|---|---|---|---|---|---|
| `SMAP-VQ-S` | 4.012 | 3.297 | 2.711 | 2.217 | 1.706 | 1.524 |
| `SMAP-VQ-B` | 3.502 | 2.706 | 1.962 | 1.743 | 1.398 | 1.279 |
| `SMAP-VQ-L` | 3.241 | 2.191 | 1.687 | 1.479 | 1.175 | 1.012 |

(b) reconstruction FID (`KL`).

| #token | 32 | 64 | 96 | 128 | 192 | 256 |
|---|---|---|---|---|---|---|
| `SMAP-KL-S` | 1.493 | 1.245 | 1.094 | 0.998 | 0.927 | 0.879 |
| `SMAP-KL-B` | 1.311 | 1.147 | 0.912 | 0.813 | 0.763 | 0.731 |
| `SMAP-KL-L` | 1.271 | 0.831 | 0.749 | 0.671 | 0.612 | 0.569 |

(c) reconstruction FID (`SoftVQ`).

| #token | 32 | 64 | 96 | 128 | 192 | 256 |
|---|---|---|---|---|---|---|
| `SMAP-SoftVQ-S` | 1.320 | 0.997 | 0.918 | 0.879 | 0.755 | 0.713 |
| `SMAP-SoftVQ-B` | 1.287 | 0.862 | 0.751 | 0.694 | 0.611 | 0.575 |
| `SMAP-SoftVQ-L` | 1.191 | 0.701 | 0.502 | 0.420 | 0.389 | 0.366 |

Table 7: **ImageNet-1K** $512 \times 512$ **generation results evaluated with ADM Dhariwal & Nichol (2021a).** †: Trained on OpenImages Kuznetsova et al. (2020) ‡: Trained on OpenImages, LAION-Aesthetics/-Humans Schuhmann et al. (2022). P: generator's parameters. S: sampling steps. T: throughput as samples per seconds on A100 with float32 precision, measured with *w/ guidance* variants if available. "guidance" refers to classifier-free guidance.

| tokenizer | rFID↓ | generator | w/o guidance gFID↓ | w/o guidance IS↑ | w/ guidance gFID↓ | w/ guidance IS↑ | P↓ | S↓ | T↑ |
|---|---|---|---|---|---|---|---|---|---|
| | | *diffusion-based generative models* | | | | | | | |
| VAE ‡ | 0.19 | UViT-L/4 Bao et al. (2023) | 18.03 | 76.9 | 4.67 | 213.3 | 287M | 50 | 1.0 |
| | | UViT-H/4 Bao et al. (2023) | 15.71 | 101.3 | 4.05 | 263.8 | 501M | 50 | 0.6 |
| | | DiT-XL/2 Peebles & Xie (2023) | 12.03 | 105.3 | 3.04 | 240.8 | 675M | 250 | 0.1 |
| | | *transformer-based generative models* | | | | | | | |
| MaskGIT-VQGAN Chang et al. (2022) | 1.97 | MaskGIT-ViT Chang et al. (2022) | 7.32 | 156.0 | - | - | **177M** | 12 | 3.9 |
| LFQ Yu et al. (2024a) | 1.22 | MAGVIT-v2 Yu et al. (2024a) | 4.61 | 192.4 | - | - | 307M | 12 | 3.5 |
| | | | **3.07** | **213.1** | 1.91 | 324.3 | 307M | 64 | 1.0 |
| SMAP-L-64 | 1.78 | MaskGIT-ViT Chang et al. (2022) | **3.64** | 179.8 | 2.74 | 221.1 | **177M** | **8** | **41.0** |
| SMAP-B-128 | 1.37 | MaskGIT-ViT Chang et al. (2022) | 3.91 | **182.0** | 2.49 | 260.4 | **177M** | **8** | 33.3 |
| | | | 4.17 | 181.0 | **2.13** | **261.2** | | 64 | 7.4 |
| | | *ours* | | | | | | | |
| SMAP(KL)-B-128 | 0.69 | CARD-B | 4.29 | 155.2 | 2.99 | 253.1 | 234M | 25 | 11.7 |
| SMAP(KL)-B-128 | 0.69 | CARD-L | 3.12 | 211.1 | 2.11 | 298.4 | 568M | 25 | 2.8 |

## C  Detailed Results of Preliminary Experiments

We summarize the detailed results of preliminary experiments in Table 6 and Table 7.

## D  Limitations

Our study has several limitations. Most importantly, we evaluate semantic-aware prefix learning only under class-conditional supervision on ImageNet. While this setting is sufficient to isolate the role of semantic conditions in tokenizer pretraining, it remains substantially simpler than text-conditioned or multimodal generation, where semantic inputs are richer and more compositional. In addition, our downstream generation experiments are centered on `CARD` as a representative generator built on top of `SMAP`. Although this is adequate to show that semantically grounded tokenization improves generation, broader validation across other generator architectures would be needed to establish full generality. Finally, we restrict our experiments to the image domain. Extending semantic-aware prefix learning to text-conditioned synthesis, multimodal conditioning, and spatiotemporal settings such as video remains an important direction for future work.

## E  Dataset Licenses

The datasets we used for training and/or testing `SMAP` are described as follows.

**ImageNet-1K:**  We train and evaluate `SMAP` on ImageNet-1K generation benchmark. This dataset spans 1000 object classes and contains 1,281,167 training images, 50,000 validation images and 100,000 test images.

We use the training set for our tokenizer and generator training. The validation set is used to compute reconstruction FID for evaluating tokenizers. The generation results are evaluated with generation FID using pre-computed statistics and scripts from ADM Dhariwal & Nichol (2021a) [1].

License: `https://image-net.org/accessagreement`

URL: `https://www.image-net.org/`

---

[1]`https://github.com/openai/guided-diffusion/tree/main/evaluations`

