# OpenReview forum: "Semantic-Aware Prefix Learning via Token Truncation for Efficient Image Generation"
_TMLR — Decision pending for TMLR_

### Review · Reviewer_qzRP · 2026-04-12

**Summary Of Contributions:**

This paper introduces SMAP, a semantic-aware 1D image tokenizer, and CARD, a hybrid generator combining autoregressive modeling with flow-matching. Addressing the limitation that most tokenizers prioritize pixel-level reconstruction over high-level meaning, the authors propose making semantic information functionally essential. By injecting semantic embeddings into the encoder and decoder and employing a tail token dropping strategy during training, SMAP forces the model to prioritize a semantic prefix. This results in information-ordered, length-adaptive latent sequences where early tokens carry the most critical structural and conceptual data. Building on this representation, the CARD generator models global structure autoregressively before refining details via flow-matching. Evaluated on ImageNet, the framework demonstrates superior reconstruction quality compared to traditional 1D tokenizers and achieves competitive class-conditional generation performance.

**Additional Comments:**

The related work could be strengthened by discussing a few additional tokenizer papers, especially GigaTok [1] and ElasticTok [2].

[1] GigaTok: Scaling Visual Tokenizers to 3 Billion Parameters for Autoregressive Image Generation. ICCV 25

[2] ElasticTok: Adaptive Tokenization for Image and Video. arXiv preprint arXiv:2410.08368 (2024).

**Audience:**

Yes

**Audience Explanation:**

This is a thoughtful and promising paper with a strong central idea. The notion that semantics should become a persistent prefix-level information pathway, rather than a weak auxiliary alignment target, is compelling. The method is simple, the empirical reconstruction gains are strong, and the generation results are competitive under compact token budgets.

**Claims And Evidence:**

Yes

**Claims Explanation:**

+ This paper is well motivated. It identifies a real limitation of prior tokenizer training pipelines: semantics are often encouraged only through loose alignment objectives rather than being made necessary for reconstruction. This framing is conceptually clear and gives the work a strong central thesis.

+ The core idea is simple and effective. Semantic injection plus suffix truncation is an elegant design, and the truncation mechanism creates a natural inductive bias toward prefix-ordered representations.

+ The reconstruction gains are strong, especially at compact token budgets. For example, compared with TiTok, SMAP substantially improves rFID in both VQ and KL settings, which supports the claim that the learned token space is more efficient.

+ The paper does not stop at reconstruction. It also evaluates downstream generation and shows that semantic-aware tokenization improves gFID, suggesting that the tokenizer improvements are useful beyond autoencoding.

**Requested Changes:**

- The main limitation is the validation scope. The paper only studies class-conditional ImageNet, where the semantic input is a class embedding. How well does semantic-aware prefix learning transfer beyond class-conditional ImageNet to richer conditioning such as text prompts or multimodal inputs?

- Could the condition embedding be acting mainly as a class-level prototype prior? Have you done any probing or retrieval on the learned condition space?

- Why is suffix truncation the right choice? Have you compared it with random token dropout?

- How much of the gain comes from improved latent representation learning versus simply giving the decoder stronger side information? For example, what happens with encoder-only or decoder-only semantic injection?

- SMAP feels like the main conceptual contribution, while CARD is more of a downstream generator built to exploit the learned latent space. How specific is the downstream benefit to CARD? Does SMAP also help when paired with more standard generators?

---

### Review · Reviewer_e1Z1 · 2026-04-13

**Summary Of Contributions:**

Strengths:
1. The idea is simple and interesting. SMAP makes high-level semantics functionally indispensable for representation learning by injecting them as prefix-preserved invariants and enforcing dependency through a tail token dropping strategy.
2. The framework achieves competitive high-fidelity reconstruction and generating using only 128 tokens.
3. The proposed method simplifies the training process into a single stage.

**Audience:**

Yes

**Audience Explanation:**

The audience at TMLR would be interested in this paper because it addresses a fundamental bottleneck in latent image modeling: the gap between semantic understanding and pixel reconstruction. By providing a simplified one-stage training recipe and demonstrating strong performance on a compact token budget , the paper offers insights into how semantic signals can be better integrated into generative pipelines.

**Broader Impact Concerns:**

I haven't found a broader impact statement. The authors may include discussions on the dataset biases.

**Claims And Evidence:**

Yes

**Claims Explanation:**

Yes, the claims are supported by evidence within the scope of ImageNet-1K class-conditional generation. The ablation studies (Table 3) and qualitative visualizations (Figure 4) provide a convincing link between the proposed token truncation strategy and the resulting semantic grounding. However, the evidence for the method's superiority in downstream generation is more marginal, and its effectiveness on more complex, compositional semantic tasks remains unproven. The details can be found in the section of requested change.

**Requested Changes:**

I would like to see more analysis on the following points.
1. Efficiency and complexity analysis of CARD
- In Table 4, the improvement in gFID over existing models like TexTok is quite marginal.
- The authors are expected to provide a comparison of FLOPs and inference latency (latency per image) for CARD vs. pure AR and Diffusion baselines. This is necessary to justify the hybrid architecture's complexity.

2. Different resolutions.
- Similar to TiTok and TexTok, elevate the 512x512 analysis from the Appendix  to the main text, with a discussion on how structural coherence is maintained at higher pixel-to-token ratios.

3. Robustness and generalization
- Evaluate the SMAP tokenizer using a standard, non-hybrid generator (e.g., DiT) to isolate the performance gains of the latent space from those of the CARD architecture.
- The class embedding idea is expected to work well on ImageNet dataset, but it is not convincing the semantic prefix mechanism works well on multi-concept semantics.

4. While the submission successfully simplifies the training pipeline into one stage, the evidence in Table 4 shows that it has not yet outperformed the gFID of more established multi-stage or distillation-based systems. Specifically, MAETok + Lightning DiT (1.73) and TexTok-256 (1.46) both achieve superior gFID scores at the same 128-token budget. The authors should clarify if the performance gap is an inherent trade-off of the one-stage training recipe or a limitation of the CARD generator.

---

### Review · Reviewer_Cjf8 · 2026-04-20

**Summary Of Contributions:**

This paper proposes SMAP, a semantic-aware 1D image tokenizer that injects class-conditional embeddings as prefix-preserved invariants into a query-based tokenization framework (building on TiTok). The key training mechanism is **tail token dropping**: during training, a random suffix of latent tokens is removed, forcing the semantic condition and early prefix tokens to carry increasing reconstruction responsibility. This yields information-ordered token sequences. The paper also introduces CARD, a hybrid causal autoregressive–diffusion generator that first models token dependencies autoregressively, then refines via flow matching. Experiments on ImageNet 256×256 and 512×512 show improved reconstruction (rFID) and competitive generation (gFID/IS) under compact 128-token budgets.

**Key strengths:**
- The tail token dropping idea is simple, elegant, and well-motivated — it makes semantic conditions functionally necessary rather than auxiliary.
- Comprehensive ablations (Table 2, 3, Figure 5) cleanly isolate contributions of each component.
- Unified framework supporting VQ, KL, and SoftVQ regularization in a single architecture.
- The cross-reconstruction visualization (Figure 4) provides compelling qualitative evidence of the semantic/detail factorization.

**Key weaknesses:**
- Evaluation is restricted to class-conditional ImageNet; generality to text-conditioned or open-domain settings is unknown.
- CARD's contribution is entangled with SMAP's — it's hard to assess how much CARD itself matters vs. the tokenizer.
- Several important experimental details and comparisons are missing (elaborated below).

**Additional Comments:**

- The paper is generally well-written and clearly structured. The progression from problem identification (Section 1) → method (Section 3) → ablations (Section 4) is logical and easy to follow.

- The naming "SMAP" and "CARD" is memorable, though "CARD" is somewhat overloaded (it could be confused with CARD in the conditional density estimation literature).

- The claim that SMAP "significantly simplifies the training pipeline in a single stage" (p.5) compared to prior multi-stage approaches is useful but under-documented. A concrete comparison of training pipeline complexity (number of stages, total GPU hours, hyperparameter sensitivity) vs. TiTok/TA-TiTok would make this claim more convincing.

- The paper would benefit from a failure case analysis: when does tail token dropping hurt? Are there image categories where the semantic condition is ambiguous or uninformative (e.g., fine-grained classes within a superclass), and does this degrade the information ordering?

- Minor: the paper cites "Sahoo et al., 2024" for MDTv2-XL/2 in Table 4, but MDTv2 is by Gao et al. This appears to be a citation error.

**Audience:**

Yes

**Audience Explanation:**

The problem of improving visual tokenizers for latent image generation is interesting. The tail token dropping strategy is a clean, general-purpose idea that could influence future tokenizer designs beyond the specific SMAP architecture. The paper sits at the intersection of representation learning and generative modeling, which is of broad interest to the TMLR community. The information-ordering property (length-adaptive encoding) has practical implications for variable-compute generation.

**Broader Impact Concerns:**

The paper proposes improvements to class-conditional image generation on ImageNet. As with all image generation work, there are potential concerns around deepfakes and synthetic media misuse. However, the method is class-conditional (not text-to-image), operates at 256×256 resolution, and is evaluated on a well-known academic benchmark. The risk profile is comparable to prior work (DiT, MAR, etc.) and does not warrant a specific broader impact statement beyond what is standard for the field.

**Claims And Evidence:**

Yes

**Claims Explanation:**

The core claim — that making semantics functionally necessary via tail token dropping improves tokenizer quality and downstream generation — is well supported by the ablation tables. Specifically:

1. **Table 2** cleanly shows SMAP's improved one-stage recipe outperforms TiTok across all token budgets and both VQ/KL settings. The improvements are substantial (e.g., rFID 7.72→3.24 for VQ-32).

2. **Table 3** isolates the effect of progressive token truncation, showing consistent (though modest) gains over SMAP without truncation. This is the right ablation to run.

3. **Figure 5** decomposes the generation improvement into tokenizer-side vs. semantic-sharing contributions — a thoughtful experimental design.

However, several aspects weaken the evidentiary strength:

- **Confounded comparisons in Table 2.** The "SMAP" column reflects *multiple* changes over TiTok simultaneously: (a) the improved one-stage training recipe, (b) semantic injection, (c) tail token dropping, and (d) architectural differences (e.g., reconstructing from mask tokens rather than latent tokens, different encoder-decoder design). Table 3 isolates (c) but the individual contribution of (a) vs. (b) vs. (d) is never separated. An ablation adding semantic injection *without* truncation to the original TiTok architecture would clarify this.

- **CARD evaluation lacks standalone baselines.** The paper pairs SMAP with CARD but never evaluates CARD on latent spaces from other tokenizers (e.g., standard KL-VAE from Stable Diffusion, or MAR's tokenizer). This makes it impossible to assess whether CARD is a strong generator in general or specifically benefits from SMAP's information ordering. Conversely, SMAP's continuous latent space is never evaluated with an existing generator like SiT or DiT (only the VQ variant is tested with LlamaGen).

- **Table 4 comparison fairness.** SMAP uses a 391M-parameter tokenizer, substantially larger than most baselines (MAR: 66M, LDM: 55M, LightningDiT: 70M). The total system parameter count (tokenizer + generator) should be reported and discussed. For example, SMAP(KL)+CARD-XL totals ~1.5B parameters, while LightningDiT is 675M+70M=745M and achieves gFID=1.35 vs. SMAP's 1.85.

- **Missing perceptual metrics.** The paper reports only FID and IS. LPIPS, PSNR, and SSIM for reconstruction, and Precision/Recall for generation, would strengthen the evaluation. FID alone can be misleading about mode coverage vs. quality tradeoffs.

- **Statistical significance.** No error bars or confidence intervals are reported for any metric. Given that FID can vary meaningfully across random seeds (especially at 50K samples), this is a notable omission.

**Requested Changes:**

**Critical (required for acceptance):**

1. **Disentangle ablation components.** Add an ablation that separates the architectural improvements (mask-token reconstruction, one-stage recipe) from the semantic injection and tail token dropping. Currently Table 2 conflates all changes. At minimum, show results for: (a) improved architecture *without* semantic injection, (b) semantic injection *without* truncation (this exists in Table 3), (c) full SMAP. This is essential to substantiate the paper's central claim that semantic injection + truncation is the key driver, not architectural changes.

2. **Cross-evaluate CARD with other tokenizers.** Train CARD on at least one standard tokenizer (e.g., the KL-VAE from SD or MAR's tokenizer) to show that SMAP's latent space specifically benefits generation, rather than CARD simply being a good generator. Alternatively, evaluate SMAP's continuous latent space with an existing generator (e.g., SiT-XL).

3. **Report total system parameters in Table 4.** The current presentation obscures the large tokenizer cost. Add a "Total Params" column and discuss the parameter-efficiency tradeoff explicitly.

4. **Clarify the training loss.** The paper never explicitly states the reconstruction loss function. Is it L1, L2, perceptual loss, GAN loss, or a combination? This is a critical reproducibility detail that is entirely missing from both the main paper and appendix.

5. **Sampling distribution for $k$.** The paper states $k \sim \text{Unif}\{0, 1, \ldots, K\}$. Has a non-uniform distribution (e.g., biased toward smaller $k$ to more aggressively force semantic responsibility) been explored? At minimum, discuss why uniform was chosen and whether the results are sensitive to this choice.

**Strongly recommended (would significantly strengthen the paper):**

6. **Add Precision/Recall metrics** for generation and LPIPS/PSNR/SSIM for reconstruction to provide a more complete picture beyond FID/IS.

7. **Report error bars** for at least the main generation results (Table 4) across multiple sampling seeds.

8. **Discuss computational overhead.** The tail token dropping requires variable-length decoding during training. What is the wall-clock training time overhead compared to standard TiTok training? The training cost in Appendix A is given in absolute terms but not compared to baselines.

9. **Notation inconsistency.** In Eq. (4), the decoder output is written with a trailing semicolon/empty slot. In Eq. (5), latent tokens appear as $\text{Regu}(Z_{1:k})$ without the "1D" superscript used elsewhere. These should be made consistent.

10. **Text-conditioned experiments.** While acknowledged as a limitation, even a small-scale experiment (e.g., on COCO with CLIP text embeddings replacing class embeddings) would substantially strengthen the generality claim. The architectural design supports arbitrary conditional embeddings $C$, so this seems feasible.

11. **Figure 3(b) is incomplete.** The comparison only includes TiTok and TA-TiTok. Adding SoftVQ-VAE (Chen et al., 2025) — whose codebase SMAP uses — as a baseline would be informative, since readers will want to know how much improvement comes from the SMAP-specific innovations vs. the underlying codebase improvements.

---

### Decision · Action_Editor_yEyQ · 2026-05-27

**Recommendation:** Accept with minor revision

**Additional Comments:**

Before publication, the authors should address several clarification and presentation issues raised by the reviewers:

- Clarify the contribution of the semantic injection and truncation mechanism relative to the accompanying architectural and training modifications through additional discussion and, where feasible, further disentangled ablations.
- Improve discussion of the relationship between the proposed tokenizer and the CARD generator, including limitations of the current evaluation setup.
- Explicitly report reconstruction loss details and improve reproducibility-related descriptions.
- Add discussion of computational efficiency, parameter counts, and inference/training overheads.
- Strengthen the presentation of higher-resolution results and discuss generalization limitations beyond ImageNet class-conditioning.
- Address minor presentation issues, missing citations, and notation inconsistencies identified by the reviewers.

While some requested experiments may be beyond the scope of a minor revision, the paper would benefit from clearer discussion of these limitations and future directions.

**Audience:**

Yes

**Audience Explanation:**

The paper addresses a timely problem in generative modeling and representation learning: how to design compact latent tokenizers that better preserve semantic structure for downstream generation. The proposed semantic-aware prefix mechanism and token truncation strategy are simple, general, and likely to influence future work on latent tokenization and adaptive representation learning.

Reviewers consistently noted that the work is technically interesting, practically motivated, and relevant to ongoing developments in autoregressive and diffusion-based image generation. The one-stage training pipeline, compact token budget, and information-ordered latent representation are all aspects likely to attract attention from researchers working on efficient generative models, tokenizer design, and multimodal representation learning.

**Claims And Evidence:**

Yes

**Claims Explanation:**

The submission presents a clear and well-motivated approach for semantic-aware image tokenization and demonstrates empirical evidence within the scope of class-conditional ImageNet generation. Multiple reviewers agreed that the core idea—making semantic information functionally indispensable through semantic prefix injection and tail token dropping—is empirically supported. The ablation studies and qualitative analyses isolate the contribution of the truncation mechanism and provide evidence for the claimed information-ordering property of the latent tokens.

The reconstruction improvements over prior 1D tokenizers at compact token budgets are consistently observed across settings. The generation experiments further demonstrate that the learned representations are useful for downstream generative modeling.

That said, several reviewers also identified limitations in the current evidentiary scope. In particular, the comparisons in the main reconstruction table conflate multiple architectural and training changes, making it difficult to fully disentangle the contribution of semantic injection versus other modifications. In addition, the interaction between the tokenizer and the proposed hybrid generator remains somewhat entangled, and broader validation beyond ImageNet class-conditioning is currently missing. Nevertheless, the existing evidence is sufficient to support the paper’s primary claims within its stated experimental setting.